



# BARRA v1.0: Kilometre-scale downscaling of an Australian regional atmospheric reanalysis over four midlatitude domains

Chun-Hsu Su[1], Nathan Eizenberg[2], Dörte Jakob[1], Paul Fox-Hughes[3], Peter Steinle[1], Christopher J. White[4,5], Charmaine Franklin[1]

[1]Bureau of Meteorology, Docklands, Victoria 3008, Australia
[2]Department of Earth Sciences, The University of Melbourne, Parkville, Victoria 3010, Australia
[3]Bureau of Meteorology, Hobart, Tasmania 7000, Australia
[4]Department of Civil and Environmental Engineering, University of Strathclyde, Glasgow, Scotland, UK
[5]School of Engineering, University of Tasmania, Hobart, Australia

*Correspondence to*: Chun-Hsu Su (chunhsu.su@bom.gov.au)

**Abstract.** The development of convection-permitting models (CPMs) in numerical weather prediction has facilitated the creation of km-scale (1-4 km) regional reanalysis and climate projections. The Bureau of Meteorology Atmospheric high-resolution Regional Reanalysis for Australia (BARRA) also aims to realise the benefits of these high-resolution models over Australian sub-regions for applications such as fire danger research, by nesting them in BARRA's 12 km regional reanalysis (BARRA-R). Four mid-latitude sub-regions are centred on Perth in Western Australia, Adelaide in South Australia, Sydney in New South Wales (NSW), and Tasmania. The resulting 29-year 1.5 km downscaled reanalyses (BARRA-C) are assessed for their added skill over BARRA-R and global reanalyses for near-surface parameters (temperature, wind and precipitation) at observation locations and against independent 5 km gridded analyses. BARRA-C demonstrates better agreement with point observations for temperature and wind, particularly in topographically complex regions and coastal regions. BARRA-C also improves upon BARRA-R in terms of intensity and timing of precipitation during the thunderstorm seasons in NSW, and spatial patterns of sub-daily rain fields during storm events. However, as a hindcast-only system, BARRA-C largely inherits the domain-averaged biases and temporal variations of biases from BARRA-R. Further, BARRA-C reflects known issues of CPMs: overestimation of heavy rain rates and rain cells, and underestimation of light rain occurrence.

## 1 Introduction

At horizontal kilometre-scales (1-4 km), convection-permitting models (CPMs) have provided a step-change in weather forecasting capabilities, particularly for forecasting rainfall and cloud cover (e.g., Lopez et al., 2009; Mailhot et al., 2010; Brousseau et al., 2016; Clark et al., 2016) and over local regions with complex terrain or land-sea boundaries (Calmet et al., 2018). Similarly, CPMs have provided new insights in regional climate projections (e.g., Argüeso et al., 2014; Prein et al., 2015; Kendon et al., 2017; 2019) over global models and major efforts are underway toward refining the horizontal resolution of climate models to km-scale (Schär et al., 2020). Extreme weather events such as thunderstorms, damaging



winds, and hailstorms, are better represented in higher resolution models (Walsh et al., 2016). General practice is that grid spacings less than about four km are required to explicitly model small convective cloud processes, replacing parameterizations of moist convection. This avoids several issues seen in parameterized convection schemes used in models

with a grid spacing greater than 10 km (Lean et al., 2008) and the "grey zone" issues in mesoscale (4-10 km) scale models (Gerard et al., 2009). A common assumption of traditional convective parameterization is that cloud fields adjust so much more rapidly than the processes forcing it that this adjustment can be modelled as instantaneous. Such schemes thus have no "memory" of the meteorological flow, leading to unrealistic model behaviours including premature convective initiation, misrepresented diurnal cycle of precipitation, over-estimation of drizzle occurrence, under-estimation of extreme rainfall

(Lean et al., 2008; Clark et al., 2016), fewer identifiable mesoscale convective systems with less structure (Done et al., 2004), and rainfall coastal locking where precipitation generated over the sea does not penetrate inland (Bureau of Meteorology, 2018). When the parameterization scheme is used at a finer resolution than 10 km, it also tends to produce intermittent on-off behaviour of deep convection (Gerard et al., 2009).

        By contrast, CPMs can represent deep convection and mesoscale convective organization explicitly on the model

grid. Explicit representation of convection improves the physical nature of precipitation persisting across orographic or land-sea boundaries by the advection of clouds/precipitation. Better representation of topography leads to improved wind circulation patterns and resulting vertical velocities (e.g., Fosser et al., 2015). Improved modelling of the interactions between storm cells and their organisations should improve the estimation of damaging winds. Many studies have found a better diurnal cycle of tropical convection over land, cloud vertical structure, and coupling between moisture convection and

convergence in CPMs (Stein et al., 2015; Leutwyler et al., 2017). A finer grid resolution can improve the flow and wind simulation over the recirculation zone behind the escarpment of a hill and higher vertical grid resolution improves simulation on the lee side of hills (Ma and Liu, 2017).

        These benefits from using CPMs are however yet to be fully realized in atmospheric reanalyses. Atmospheric reanalyses combine prior knowledge of physical processes captured in the models with observations from a diverse range of

instruments, to form spatially complete representations of the historical atmospheric conditions. They are therefore invaluable for revisiting the local processes, climate signals or events that were not fully observed, for applications such as climate monitoring and change assessments (Kendon et al., 2017; 2019), renewable energy assessment (e.g., Frank et al., 2020), and hazard management (e.g., Vitolo et al., 2019). Global-scale reanalyses have advanced in quality and quantity during the past three decades with improvements to models, data assimilation methods, number of observations and

ensemble methods (Kalnay et al., 1996; Ebita et al., 2011; Gelaro et al., 2017; Dee et al., 2011), and with increasing spatial resolutions. The latest addition, ERA5 (Hersbach et al., 2020), has a horizontal spacing of 31 km. Users of reanalyses have called for development towards finer spatial and temporal scales, below 10 km horizontal spacing and sub-daily time intervals (Gregow et al., 2016). Higher-resolution reanalysis is needed in localized climate monitoring where local-scale mechanisms influenced by complex topography, coastlines and convective processes are responsible for local climate

features and feedbacks.





Departing markedly from the global reanalyses are the regional reanalyses that use limited area models at higher horizontal resolutions over sub-regions such as North America (Mesinger et al., 2006), the Arctic polar region (Bromwich et al., 2016), Europe (Borsche et al., 2015 and references therein), India (Mahmood et al., 2018) and Australia (Su et al., 2019). These reanalyses use grid lengths in the order of 10 km to improve the representation of sub-daily variability and near-surface weather. These are generally produced with global atmosphere model configurations that include convection parameterizations (e.g., Su et al., 2019). Recently, Wahl et al. (2017) overcame this with a 7-year 2 km reanalysis over Germany with assimilation of conventional observations and radar-derived rain rates and demonstrated improved spatiotemporal variability and intensity frequency of precipitation. Such a direction in the development of the reanalyses, combined with the higher resolution regional projections, can offer a more accurate picture of changes in regional meteorology and extreme weather in the changing climate.

Dynamical downscaling is frequently used to estimate the dynamic variables at scales finer than coarser-resolution climate or weather models. This approach is undertaken at the Bureau of Meteorology (Bureau) in Australia to produce km-scale weather forecasts and/or ensemble forecasts over major cities and a 1.5 km forecast-only model has been used since 2017 for added value over the Bureau's lower resolution global system. This goal is also pursued in the Bureau of Meteorology Atmospheric high-resolution Regional Reanalysis for Australia (BARRA; Jakob et al., 2017) project. Within this context, this paper is a companion paper to Su et al. (2019) where an Australian regional 12 km reanalysis system (BARRA-R) was presented. Here we describe dynamical downscaling of BARRA-R using the UK Met Office (UKMO) Unified Model (UM) at a 1.5 km horizontal grid length over four mid-latitude sub-regions of Australia (Figure 1) over 29 years from January 1990 to February 2019. These regions are chosen in partnership with state fire and emergency management agencies, because of the important advantages dynamically downscaled reanalyses can provide for local-scale planning and management to reduce future risks due to extreme weather events such as bushfires. The four downscaling models, collectively referred to as BARRA-C, yield gridded products that include a variety of 10 minutes to hourly surface parameters describing weather and land-surface conditions and hourly upper-air parameters covering the troposphere and stratosphere with a 40 km model top on 70 model levels and 37 pressure levels.

This paper describes the model and the experimental design in Sect. 2, and Sect. 3 provides the first assessment of the downscaled reanalysis with focus on screen-level temperature, 10 m wind and precipitation. Comparisons with the BARRA-R and global reanalyses are also made to illustrate the added value of BARRA-C. Our findings are further discussed in Sect. 4, with an overall summary in Sect. 5.

## 2 BARRA-C

The development of BARRA is based on the Bureau's operational deterministic NWP forecasting over the Australian region using the Australian Community Climate and Earth-System Simulator systems ACCESS-R and ACCESS-C (Puri et al., 2013). The operational version at the time (Australian Parallel Suite 2) of ACCESS-R is the national 12 km 6-hourly





analysis/assimilation and 3-day forecasting system (Bureau of Meteorology, 2016). It has provided the initial and boundary conditions to initialize and constrain ACCESS-C over 6 smaller domains centred at the Australian cities up till 2020 (Bureau of Meteorology, 2018). The APS2 ACCESS-C dynamically downscales ACCESS-R to provide 6 hourly, 1.5-day forecasts at 1.5 km horizontal resolution. The relation between BARRA-R and BARRA-C mirrors this, but is implemented with shorter forecast (or hindcast) range, a newer version of the meteorological forecast model and science configuration (Section 2.1). In particular, BARRA-R is nested in ERA-Interim reanalysis (Dee et al., 2011) and includes 4 assimilation cycles per day (Su et al., 2019), and BARRA-C is a hindcast-only system. While BARRA-C refers to the collection of the four sub-domain models, we use BARRA-AD, BARRA-PH, BARRA-SY and BARRA-TA to denote individual domains centred at Adelaide (South Australia, AD), Perth (Western Australia, PH), Sydney (New South Wales, SY), and Tasmania (TA) (Figure 1).

The PH and AD domains are similar in terms of climate, with arid deserts north of their domains, and temperate dry hot or warm summers near coasts, and arid steppe climate in-between (Peel et al., 2007). SY has a temperate climate with warm to hot summer and lacks a dry season, while TA differs with cooler summer. Cool-season (C3) perennial grass is the dominant vegetation over the southwestern region of PH and near-coast region of AD, and broadleaf trees are widespread in the SY and TA domains (Figure S1 in the Supplement). There are several large ephemeral salt lakes (e.g., Lake Torrens, Lake Gairdner) in the AD domain, and these are modelled as land points with bare soil. It is only SY that has a distinct thunderstorm season during November-March. Thunderstorms are far less frequent in the other three domains due to lower incidence of warm, humid air masses favourable for storm development and stable conditions during the potentially favourable warmer months owing to subtropical high-pressure belt over or near these areas (Kuleshov et al., 2002). The SY and TA domains are topographically complex, with the Great Dividing Range extending north to south through the SY domain and landscape of plateaus and low mountain ranges in the TA domain.

## 2.1 Forecast model

The UM (Davies et al., 2005; version 10.6) is the grid-point atmospheric model used in BARRA and ACCESS. It uses a non-hydrostatic, fully compressible, deep atmosphere formulation and its dynamical core (Even Newer Dynamics for General atmospheric modelling of the environment, ENDGame) solves the equations of motion using mass-conserving, semi-implicit, semi-Lagrangian (SL), time integration methods (Wood et al., 2014). The prognostic variables are three-dimensional wind components, virtual dry potential temperature and Exner pressure, dry density, and mixing ratios of moist quantities. The model in BARRA-C is run with a 60-second time step, and is discretized horizontally on a regular Gaussian $0.0135° \times 0.0135°$ (about 1.5 km at the equator) grid with Arakawa-C staggering (Arakawa and Lamb, 1977), and vertically with the Charney–Phillips staggered 70-level grid (Charney and Phillips, 1953). Here the vertical levels follow model orography at the surface and relax to surfaces of uniform radial height after 50 model levels in the upper atmosphere with the model top height of 40 km.

Table 1 summarizes the differences between BARRA-C, BARRA-R, and the first UM Regional Atmosphere and Land (RAL) configuration of Bush et al. (2020). In particular, BARRA-R is based on Global Atmosphere (GA6)





configuration of Walters et al. (2017), while BARRA-C is based on the UK Met Office operational suite OS36, which preceded the release of RAL1. The physical parameterization schemes common to BARRA-C and BARRA-R include a variant of Wilson and Ballard (1999) for mixed-phase cloud microphysics, the large-scale cloud scheme of Smith (1990), and the radiation scheme of Edwards and Slingo (1996), all of which have been improved since publication. The UM uses a

convection parameterization scheme based on Gregory and Rowntree (1990), which is not used in BARRA-C. At the grid length of 1.5 km, the horizontal grid length approaches the depth of the boundary layer (Hanley et al., 2015) and as such it is no longer appropriate to use the 1D boundary layer parameterization that restricts mixing to the vertical. BARRA-C therefore uses a blended boundary layer parameterisation (Boutle et al. 2014) where the scheme transitions from the 1D vertical turbulence scheme of Lock et al. (2000) to a 3D subgrid turbulence scheme based on Smagorinsky (1963) as a

function of the grid length to the turbulent length scale. The mixing length of 300 m, which can be tuned to control the smoothness of the fields and the number of small cells, is taken from the operational systems.

The cloud scheme uses a profile of critical relative humidity values (RHcrit), above which a grid box contains some cloud if the relative humidity is exceeded. Based on the assumption that there should be less subgrid variability in humidity in smaller grid boxes, BARRA-C uses higher RHcrit values that BARRA-R in the lowest few layers, decreasing smoothly

above to 0.8.

Without the convection parameterization scheme, BARRA-C relies on the model dynamics to represent convective motions. While convection remains unresolved in 1.5 km models, removal of the cumulus parameterization has shown to result in more realistic behaviour (Clark et al., 2016). In particular, the model can explicitly capture processes with convective-like characteristics, which can subsequently drive scales that the model can properly resolve. BARRA-C also

reduces the appearance of unrealistically strong vertical velocities and "grid-point storms" seen in BARRA-R due to the inability of convective parameterization to stabilize the air column (Su et al., 2019). Nevertheless, convection can remain under-resolved, leading to cases of too-early small, shallow showers or none at all. The mid-latitude version of RAL1 therefore includes stochastic perturbations of temperature and moisture and relative weak turbulent mixing, to encourage the model fields to be less uniform and help convection to initiate. It is of note that the stochastic perturbations of moisture are

absent in BARRA-C.

Another distinguishing feature of BARRA-C is the handling of mass conservation during the advection of moisture prognostic variables. BARRA-C and RAL1 use the zero-lateral flux scheme of Zerroukat and Shipway (2017) for moisture conservation at the model's lateral boundaries, avoiding spurious extreme precipitation caused by the SL treatment of moisture variables near partially-resolved convection.

BARRA uses the land surface scheme of Best et al. (2011), implemented in the Joint UK Land Environment and Simulator (JULES). It describes a 3 m four-layer soil column, with sub-surface temperature updated using a heat diffusion equation, the vertical moisture flux is estimated using the Richard's equation and Darcy's law. The soil hydraulics is computed using van Genuchten equation. It uses a nine-tile approach to represent subgrid-scale heterogeneity in land cover, with the surface of each land point subdivided into five vegetation types (broadleaf trees, needle-leaved trees, temperate



cool-season (C3) grass, tropical warm-season (C4) grass, and shrubs) and four non-vegetated surface types (urban, inland water, bare soil, and land ice). In particular, the urban surfaces are represented only by a single urban tile, where street canyons and roofs are not distinguished.

        The characteristics of the lower boundary, climatological fields, and natural and anthropogenic emissions are specified using static ancillary fields. These are created as per Bush et al. (2020; Table A1), with the exceptions of ancillaries

for the land–sea mask, canopy tree heights, and land usage. The land–sea mask is created from the 1 km resolution International Geosphere–Biosphere Programme (IGBP) land cover data (Loveland et al., 2000) for SY and TA, and from Shuttle Radar Topography Mission (SRTM) orography data for AD and PH. The canopy tree heights are derived from satellite light detection and ranging (lidar; Simard et al., 2011; Dharssi et al., 2015). The land usage ancillary, created from IGBP, is modified for AD and PH to match the water fractions in the Water Observations from Space (WOfS, Mueller et al.,

2016). Aerosol absorption and scattering in the radiation scheme assume climatological aerosol properties. A climatological ozone field is also prescribed.

## 2.2 Initial and boundary conditions

The production of BARRA-C is highly parallelized, owing to the fact that its model hindcast is re-initialized with 6-hourly initial conditions at the synoptic hours $t_0$ = 00:00, 06:00, 12:00, and 18:00 UTC created by downscaling from BARRA-R

(Figure S2 in the Supplement). These fields are taken from the centre of BARRA-R's 6-hour analysis windows. A two-component reconfiguration approach is taken, in which BARRA-R winds, moisture and temperature are downscaled separately with different resolution topography sets, to remove model instability over high topography. BARRA-C is further constrained by BARRA-R at the lateral boundaries without nudging, based on the prescription described in Bush et al. (2020) and a boundary rim width of 0.34°. The boundary conditions force the development of the larger-scale features within

the BARRA-C domains. These setups follow the Bureau's NWP system, and ensure that the benefits of analysis in BARRA-R is inherited by BARRA-C, where BARRA-C is treated as a physical interpolator of BARRA-R.

        The JULES soil moisture and temperature are prescribed by BARRA-R. Consistent with BARRA-R, daily sea surface temperature and sea ice 0.05 x 0.05° analysis from reprocessed (1985–2007; Roberts-Jones et al., 2012) and near-real-time Operational Sea Surface Temperature and Ice Analysis (OSTIA; Donlon et al., 2012) are used as lower boundaries

over the water after being interpolated to the UM grid. The NRT data are used from January 2007.

        Each hindcast in BARRA-C is a 9-hour simulation, where the model data during the first 3-hour period is discarded as the fine detail is only partially established due to model spin-up from the coarse-resolution initial conditions. In other words, the hindcast fields between $t_0$+4h and $t_0$+9h form the BARRA-C data sets. Such a hindcast length is considered short, and is chosen to meet computational constraints and regular reinitialization is needed for running the model for such an

extended period. One clear limitation of our setup is that model spin up artefacts are expected to be still present, particularly for convective clouds and rain.





## 3 Assessment

Our assessment focuses on near-surface variables and precipitation as the aim of BARRA-C is to capture small-scale local weather phenomena that is most apparent near the surface. BARRA-C hindcasts are evaluated against point-scale station observations for screen-level temperature, 10 m wind speed and precipitation. They are also compared with gridded daily analyses of these observations for temperature and precipitation. Added skills in BARRA-C are illustrated by comparing against BARRA-R, ERA-Interim and MERRA2 hindcasts, and against ERA5 hindcasts for precipitation, and ERA5 hourly analysis for the other variables. Lastly, a scale-selective evaluation of extreme storms is conducted using radar observations available over the SY domain.

## 3.1 Point evaluation of screen temperature, 10 m wind speed, surface pressure

The $t_0+6h$ model hindcasts of screen-level temperature, 10 m wind speed, and surface pressure are evaluated against land observations during the 2010-2012 period, following the approach of Su et al. (2019). These observations have no direct relation to BARRA-C, since there is no analysis in BARRA-C and they are not used in the associated BARRA-R cycle $t_0$. These fields are interpolated from the model levels using surface similarity theory (Walters et al., 2017). Our benchmarks include BARRA-R and ERA-Interim $t_0+6h$ hindcasts, the MERRA2 hourly time-averaged hindcast fields (M2T1NXSLV), and the ERA5 hourly analysis. The models are interpolated to be coincident with the observed locations and times. As the observations are irregularly distributed in time, all observations within a $t_0+5h$ to $t_0+7h$ time window for $t_0 = 00$ and 12 UTC are considered. Root mean square difference (RMSD), Pearson's linear correlation, additive bias, and variance bias are calculated at each station between observed ($d_o$) and model ($d_m$) data, with bias = mean($d_m$) - mean($d_o$) and the variance bias as Mbias = var($d_m$)/var($d_o$)-1, to capture differences in the dispersion, where var(*) computes the variance in time. This assessment does not serve to inform the true quality of the various reanalyses at their native resolutions; it indicates whether the models contain finer-scale information captured by point measurements. Based on Di Luca et al., (2016), we distinguish three distinct regions with characteristics of complex topography (stations with an elevation higher than 500 m – *topo*), land-sea contrasts (stations that are within 1.5⁰ of the coast – *coast*), or a relatively smooth terrain (stations far from the coast – *flat*) (Figure S3 in the Supplement).

The comparisons of scores across all BARRA-C domains are shown in Figure 2. For temperature, the BARRA (i.e., BARRA-R and BARRA-C) and ERA5 show better agreement with the station data than the other coarser reanalyses for most metrics. For instance, BARRA-C shows lower RMSD than ERA-Interim at 80% of stations. BARRA shows greater contrast from the global reanalyses than between them. ERA5 shows warm (additive) bias, while the BARRA appears cooler. ERA-Interim and ERA5 generally show less variability in temperature than observations (Mbias < 0) while the other models tends to have more similar temperature variability with observations. This is related to the cold bias in ERA during high temperature (shown in the next section). On average, BARRA scores lower RMSD than ERA5 at elevated stations (e.g., Snowy Mountains in SY) and smaller Mbias at near-coast stations. Similarly, BARRA-C shows more visible improvements





to BARRA-R at stations near coasts or over complex topography in terms of RMSD, correlation and Mbias (Figure S3 in the
Supplement). Consequently, BARRA-TA scores higher than BARRA-R on average. BARRA-C shows higher RMSD in the
flat regions than in the other regions, unlike the other reanalyses. The degradation is small, within 0.6 K in terms of RMSD,
and for AD, this is related to over-dispersion (MBias > 1).

For 10 m wind speed, BARRA-C, BARRA-R and ERA5 similarly exhibit lower RMSD and higher correlation with
the station data than the other global reanalyses, and the differences between these three models are not pronounced.
BARRA's largest enhancement to ERA-Interim is found at elevated stations and near coasts, benefitting Tasmania
specifically. Contrasting BARRA-R, BARRA-C tends to show lower RMSD at these stations (Figure S3, Supplement), and
where we observe higher RMSD in BARRA-C, the difference is within 1 m/s. The wind estimated by all the models tends to
be under-dispersed (MBias < 1), relating to positive (negative) bias during low (high) wind conditions. Such a model under-
dispersion is more striking in the TA and SY domains than in the other domains, and over coastal regions.

For the surface pressure, the higher resolution models including ERA5 show markedly lower RMSD near coasts.
There is very good agreement between ERA5 and the observations. Some improvements to BARRA-R from BARRA-C are
mainly in correlation and Mbias and over coastal regions and mountains.

### 3.2 Comparison with gridded analysis of daily maximum and minimum screen temperature

The reanalyses are compared against a gridded daily $0.05° \times 0.05°$ analysis of observed maximum and minimum screen
temperature from the Australian Water Availability Project (AWAP; Jones et al., 2009) in Figure 3. BARRA outperforms the
driving model ERA-Interim in reducing the cold (warm) bias during summer DJF (winter JJA), particularly over the SY and
TA domains. BARRA avoids the bias in ERA over the salt lakes in SA, by modelling them as land points. BARRA-C shows
smaller extent of summer cold bias in daily maximum temperature over the Great Dividing Range than BARRA-R and
ERA5, but shares similar bias with BARRA-R elsewhere. The warm bias in daily minimum temperature in winter is also
similar between BARRA-C and BARRA-R. BARRA-C has largely inherited the biases from BARRA-R, but with small
local-scale differences. Despite such similarities in summer bias, when comparing the number of hot days exceeding 35 ℃
(308.15 K) in Figure 3(c), there are more hot days in BARRA-C than in BARRA-R over inland Australia. By contrast, the
summer cold temperature bias in both ERA reanalyses is also reflected by fewer hot days, and vice versa for MERRA2.

Figure 4 examines the inter-seasonal and inter-annual variations in temperature bias with respect to AWAP. They
are similar between BARRA-C and BARRA-R. The inter-seasonal range of bias in BARRA is around 2 K, which is similar
to ERA-Interim and MERRA2 but larger than ERA5. For AD and PH, the daily maximum temperature is positively biased
during summer months (DJF) and is negatively biased during winter (JJA). The negative bias in daily maximum temperature
is smallest during summer for SY and TA, and is largest during winter for SY. For daily minimum temperature, these are
reversed, e.g., the associated positive bias peaks during winter for AD, PH and SY, and the negative bias is maximum during
summer for AD and PH.





The inter-annual variability and trend of the bias do exist in BARRA. For daily maximum temperature bias, there is a cooling trend in AD and PH, and a warming trend in TA. These trends can also be seen in ERA5 and MERRA2. For daily minimum temperature bias, trends in BARRA are less apparent than in ERA5 and MERRA2. Here we also observe in TA that BARRA shows a small warming trend.

The distributions of daily maximum summer temperature from the reanalyses can differ significantly locally, and Figure 5 illustrates this for four AWAP grid cells near the four state capital cities. The closest model grid cells are selected, and due to differences in spatial resolution, not all models treat these cells as land points, and even as a land point, they are treated with different land cover fractions. Here, we find that BARRA-C simulates different extremes from BARRA-R for all locations but Sydney. BARRA-C tends to simulate higher temperature extremes, and in Adelaide and Tasmania, show

better agreement with AWAP than BARRA-R and the ERA reanalyses. BARRA has too many too warm days in Perth, which is also observed in Australia's western seaboard in Figure 3(c).

### 3.3 Comparison with raingauges over Sydney

Hourly modelled precipitation from BARRA and ERA5 are compared against observations from 27 raingauges within $1^{\circ}$

radius around Sydney during the warmer months (NDJF) in 2008-2013 in Figure 6. During these months, convection processes dominate and can produce a distinct diurnal distribution in thunderstorm activity, with the greatest frequency of severe thunderstorms occurring in November and December (Griffiths, et al., 1993). BARRA-R and ERA5 both underestimate the frequency of heavy rain rate > 8 mm/h, with a lesser extent for BARRA-R. By contrast, BARRA-C underestimates the frequency of light rain rate and overestimates heavy rates. BARRA and ERA5 also distribute rainfall

differently over a day. BARRA-C shows a bimodal distribution similar to the observations, albeit showing too much rain leading up to the 06 UTC peak and too little rain during the daily minimum around 18 UTC. The more pronounced diurnal cycle in precipitation is consistent with the over and under-estimation of different rain rates. BARRA-R shows less diurnal variation in rainfall with too much rain distributed during 00-06 UTC, whereas ERA5 shows a pronounced early timing bias.

### 3.4 Comparison with daily rainfall analysis

Figure 7(a) compares the modelled precipitation against daily raingauge analysis from AWAP, including MERRA2's hourly time-averaged precipitation (PRECTOTCORR) product (M2T1NXFLX). BARRA-C shows a wet bias over the Great Dividing Range and the southeast area of the AD domain, but improves the dry bias in BARRA-R and ERA reanalyses over the eastern and western seaboards, and the Fleurieu and Yorke Peninsulas of South Australia. BARRA-C also shows dry biases on the western borders of the AD and SY domains possibly due to inconsistencies with the zero-lateral moisture mass

flux boundary condition (Sect. 2.1). A striking difference between BARRA and the global reanalyses is over western Tasmania where the latter displays a dry bias.





Next in Figure 7(b), BARRA-R, ERA5 and ERA-Interim show too few heavy rain days (> 10 mm/day) over the coastlines, SA peninsula, and western Tasmania. BARRA-C improves on this, but generally simulates more heavy rain days than other reanalyses, and too few moderate-light rain days (<10 mm/day, not shown) in all domains. BARRA-R and MERRA2 generally show too many light rain days, and ERA reanalyses show too many light rain days in SY and eastern Tasmania, and too few in AD, PH, and western Tasmania.


The inter-seasonal and inter-annual variations in precipitation bias with respect to AWAP are plotted in Figure 8. As with temperature (Figure 4), they are similar between the BARRA-R and BARRA-C, although the latter shows a larger range in all but TA domains. In particular, the wet bias is generally observed during the wet season (JJA for AD, DJF for

PH), wetter months (JJA for TA) or thunderstorm season (DJF for SY), and the dry bias generally occurs during the dry season or drier months (e.g., SON for AD, PH and TA). This is consistent with the tendency of BARRA-C to overestimate heavy rain rates and underestimate light rain occurrence. Some of the inter-annual variations in the bias are clearly common amongst BARRA and the global reanalyses; examples are in AD and PH where the various models are drier during the Millennium drought (1996-2009). BARRA can also show different trends. For instance, there is a wetting trend post-2009

for BARRA in AD, but this is opposite for the other models. In SY, BARRA also displays a wetting trend, while ERA trends drier.

It should however be noted that, as is often found for gridded interpolated data, AWAP tends to underestimate the intensity of extreme rainfall events, and overestimate the frequency and intensity of low rainfall events (King et al., 2013). The errors are larger at high elevations (SY and TA) where gauges are fewer, and when there is frozen precipitation, and/or

topography is exposed to prevailing winds (Chubb et al., 2016).

### 3.5 Storms over Sydney

The point gauge-based assessment in Sect. 3.3 is harsher to higher resolution grids due to the compound error of space and time near-misses increasing as the grid cells shrink. Therefore, we compare the simulated rain fields from BARRA-SY with the Bureau's radar nowcasting rainfall product (Rainfields2; Seed et al., 2007) using fractions skill scores (FSS) to allow

assessment at different spatial scales, following the approach described in Roberts and Lean (2007), Jermey and Renshaw (2016) and Acharya et al. (2020). The FSS provides an evaluation of the rainfall skill as a function of spatial resolution. The radar product, blended with the gauge observations using conditional merging (Sinclair and Pegram, 2005), is available from 2014-onwards on a mosaic grid consisting of the domains of multiple radars. Following Acharya et al. (2020), the largest 36 storm events during 2014-2016 are selected based on domain-averaged daily precipitation.


FSS is categorised as a 'neighbourhood verification' metric (Ebert, 2009) in which fractional coverages of grid cells close to observation are valued equally. The FSS tallies the relative number of 'hits' between the model and the observation at different spatial scales and different rain thresholds. An FSS of 1 represents a perfect forecast where the number of cells with precipitation above a threshold within a neighbourhood is identical between the model and observation grids for all possible neighbourhoods. Here, BARRA hourly rain rates are regridded to the radar grid of 1.5 km, and the accumulated rain





amounts over moving 6-hour windows are analysed. From the 36 multi-day storm event set, 1323 different 6-hour events are produced using a moving window. FSS is computed for each 6-hour event for each model and then the scores are aggregated to give an average for all events. Given that inherent bias between the observation and both of the models exist due to their representativity differences and also to focus on the spatial accuracy of the models, we use percentile-based thresholds computed across all the storm events. This ensures that the model and observed rain fields have an identical fraction of rain events for each threshold value. Figure 9 illustrates the striking differences between the BARRA-R and BARRA-SY for five events in 2014. BARRA-SY can show more realistic organisation in the 1.5 km model as a consequence of the convective rain being modelled explicitly and can produce higher rainfall. The event on 7 December 2014 in Figure 9(v) illustrates a case where BARRA-R shows excessive grid-point precipitation over the mountains, which are absent in observation and BARRA-SY. At the same time, BARRA-SY can show too many cells (Figure 9(ii)), which can produce streaks of light rainfall (Figure 9(iv)).

The FSS results in Figure 10 shows that BARRA-SY is more skilful over all scales than BARRA-R for all threshold levels. $FSS_{uniform}$ is the FSS of a forecast field with a uniform fractional coverage equal to the fraction of points observed with any rain (>0.2mm/hr). Scores greater than $FSS_{uniform}$ is considered skilful. For the lowest threshold (56%, i.e. 4 mm in the observed radar values), the uniform score ($FSS_{uniform}$) is reached at scales of 0.3⁰ (BARRA-SY), and 0.65⁰ (BARRA-R). At the highest threshold (99.9%, 64 mm), the uniform score is reached at scales of 2.4⁰ and 3.35⁰, respectively. The contrast between the two BARRA FSS is therefore greater at the higher precipitation thresholds. FSS is also generally lower as the area of rain being sampled become more localized and is more challenging to be reproduced in the models.

## 4 Discussion

The dynamical downscaling of BARRA's 12 km reanalysis, BARRA-R, with the BARRA-C 1.5 km models has been shown to provide additional information about local near-surface meteorological conditions. BARRA-C provides better representative point-scale estimates of screen temperature, 10 m wind speed and surface pressure at some areas with complex topography or near coastlines, and mainly inherits the skills of BARRA-R over other areas. The degradation from BARRA-R is slight, within (RMSD) 0.6 K for temperature and 1 m/s for wind speed.

BARRA-C also shows a 2 m wind speed bias that is positive (negative) bias during light (strong) wind conditions, similar to BARRA-R. Many factors such as boundary layer mixing, form drag for subgrid orography and surface properties can influence wind estimation over land. The representation of the stable boundary layer remains challenging due to the multiplicity of physical processes (including turbulence, radiation, land surface coupling and heterogeneity, turbulent orographic form drag) involved and their complex interactions, such that models typically suffer biases in 2 m temperature and wind speed under such conditions (Steeneveld, 2014 and reference therein).

BARRA-C also inherits the domain-averaged biases in daily maximum and minimum temperature from BARRA-R. It reduces some bias over the Great Dividing Range but simulates more hot days particularly over inland Australia. The bias



varies between the four domains, with AD and PH showing a change of sign in bias between summer and winter months, while SY and TA show persisting negative (positive) bias for daily maximum (minimum) temperatures. Such similarities between the domains may be related to their similarities in terms of climate and land cover. Bush et al. (2020) discussed that

changes in land use mapping to improve the amount of vegetation cover, scalar roughness length for grass tiles, and albedo of vegetated tiles are important to improve the diurnal biases in pre-RAL1 configurations. These could benefit the biases seen over vegetated areas, particularly for daily minimum temperature in SY and TA.

The reduced dry bias of higher rain rates seen in the coarser scale models during the thunderstorm seasons in SY is alleviated by BARRA-C. The underestimation of the peak rain rates in BARRA-R and ERA5 was expected from the lack of

convection organisation due to the use of a cumulus parameterisation, whereas BARRA-C evidently shows more realistic organization. However, the latter also exhibits too much heavy rain and not enough light rain, likely due to the under resolved convection and the model's inability to resolve detrainment from convective updrafts. This is consistent with the findings reported in other studies; Lean et al. (2008) and Hanley et al. (2016) found that 1 km grid length UM simulations tend to produce cells that were too intense, too far apart and with not enough light rain. The latter also noted insufficient

small storms in both shower cases and large storm cases, and too many large cells in shower cases.

The short hindcast length in BARRA-C (Sect. 2.2) poses a further limitation. The rainfall excess could result from model spin-up, as extra CAPE builds up during the early timesteps when there is insufficient convection, which is then released (Lean et al., 2008). Champion and Hodges (2014) have also noted that modelled precipitation intensities are most accurate when the model is initialised 12 hours before the rain maxima. The moisture conserving zero-lateral mass flux

boundary conditions in BARRA-C exacerbate this issue. Moisture variables are not advected across boundaries and instead allowed to develop via physical processes in the model. These processes take some time to spin-up in each hindcast leading to near-boundary downstream moisture bias, for example, the western boundary of the annual rainfall maps of AD and SY domains (Figure 7(a)). These issues of precipitation with short hindcasts can be improved with an assimilation system that will allow high resolution features to propagate from one hindcast cycle to the next (Dixon et al., 2008). In spite of these

limitations, we find that BARRA-C provides a more representative rainfall climatology for heavy rain days near the coastal or mountainous regions, and as well as sub-daily rain spatial patterns.

BARRA-C simulates peaks in the diurnal distribution of precipitation better than BARRA-R and ERA5. However, we also find that precipitation may be initiated too early and grow too rapidly. This is contrary to the expectation for all models to initiate too late since subgrid-scale initial plumes cannot be represented. The early initiation bias in BARRA-R is due to

the CAPE-based trigger mechanism of the convection scheme (Lean et al., 2008). In the case of the km-scale UM, the reasons are likely several. Hanley et al. (2015) partly attributed timing bias in convection initiation, which is too early in shower cases and too late in the larger storm cases, to unresolved convection at the km-scale grid length. Other reasons may be that stochastic perturbations (Sect. 2.1) or model response to the pre-convective profile is too strong, or the profile has inadequate convective inhibition (CIN). The various aspects (intensity, size and timing) of simulated cells have shown to





improve with adjustments to the mixing length used in the subgrid turbulence scheme, but not all aspects improve simultaneously (Hanley et al., 2015).

There are trends and/or inter-annual variability of bias in BARRA against analyses of temperature and precipitation observations, and some of these trends are also apparent in the global reanalyses. BARRA-C largely mirrors the variability in BARRA-R, and its magnitude is of the order similar or less than the global reanalyses. Spurious trends or artificial shifts in

reanalyses could result from abrupt changes to the amount of satellite data assimilated at the start and end of satellite missions and the various observational data archives. In BARRA-R, corrections were also made to the observation screening and thinning rules mid-production (Su et al., 2019). It is however outside of the scope of this work to assess the impacts of various observational changes.

BARRA-C shows better agreement with the pattern and the relative distribution of radar-derived rainfall during storms

over Sydney, owing to the use of explicit convection (Sect 2.1) and a higher resolution model and this is consistent with earlier studies with UM (e.g., Lean et al., 2008). Comparisons of FSS from the same events including ERA5 show that the low resolution leads to significant representation errors and lower FSS than BARRA-R despite both parametrising convection (Figure S4, the Supplement). While BARRA-C still shows considerable bias compared to both rain gauges and radar observations, BARRA-C adds value to BARRA-R and ERA by providing more realistic and accurate spatial

representations of rainfall during storms at various spatial scales and percentile thresholds.

## 5 Conclusions

The recent development of CPMs in NWP has facilitated the creation of km-scale regional reanalysis and climate projections. BARRA is the first regional reanalysis that focuses on the Australasian region, which has been developed with significant co-investment from state-level emergency service agencies across Australia. BARRA-C is the critical component

of the project that provides these agencies with the means for developing a deeper understanding of past extreme weather at local scales, especially in areas that were not adequately served by observation networks (e.g., Figure S3, the Supplement). The four mid-latitude domains of BARRA-C are designed to address these needs, and BARRA-R is needed to establish a driving model for BARRA-C that is of higher resolution than ERA-Interim, and to utilise more of the Australian local observations (Su et al., 2019). Completed in June 2019, the 29-year BARRA-R reanalysis (1990 to February 2019) and its

downscaled counterparts BARRA-C form a collection of high-resolution gridded meteorological datasets with 12 and 1.5 km horizontal grid lengths and 10 minutes to hourly time resolution, produced using systems closely related to the Bureau's present (as of October 2020) regional NWP systems. The hybrid model-level and pressure-level grids from BARRA-C are also available to drive/force sub-km weather or non-weather models.

This paper describes the experimental configuration of BARRA-C and provides a preliminary assessment to

illustrate its skills over BARRA-R and the global reanalyses at their subgrid scales. As expected from a hindcast-only system, it inherits the domain-averaged biases from BARRA-R. However, there exists added skill at the local-scale for





temperature and wind, particularly in topographically complex regions in SY and TA, and coastal regions in all domains. As expected, the contrasts in skills and biases are most apparent between BARRA and the coarser-scale reanalyses (ERA-Interim, MERRA2). The BARRA-R and BARRA-C produce more distinctive precipitation estimates for intensity, sub-daily

timing and hourly spatial patterns that are characteristics of their physical schemes. BARRA-C also provides different spatial distribution of precipitation over complex terrains and more skilful representations of sub-daily rainfall fields. The latter suggests that BARRA-C is more suited for studies of extreme rainfall events, albeit high rainfall bias exists. More importantly, BARRA-R and BARRA-C can be used conjunctively to improve individual estimates of temperature and precipitation. Some of their biases, including for 10 m wind, could also be addressed via post-processing using multi-variate

regression models or quantile matching methods such as those of Glahn and Lowry (1972), and Cattoën et al. (2020). Users of BARRA are strongly encouraged to undertake a local evaluation to ascertain the skills of BARRA-C for their regions and parameters of interest.

BARRA lays some of the important groundwork for future reanalysis-related activities and developing national climate risk services at the Bureau. Some of the issues identified in this work are being actively researched by collaborating

national meteorological centres and academic institutions, within the "Regional Atmosphere" configuration development framework (Bush et al., 2020). Future reanalyses will also benefit from the recent advances in Bureau's NWP, whereby an assimilation system (Rennie et al., 2020) and ensemble are introduced in its upcoming km-scale models, to allow propagation of high-resolution information between hindcast cycles and estimation of uncertainties.

**Code availability**

All code, including the UM (version 10.6) and JULES (version 4.7), used to produced BARRA-C is version-controlled under the Met Office Science Repository Service. UM is available for use under license, http://www.metoffice.gov.uk/research/modelling-systems/unified-model. JULES is available under licence free of charge, http://jules-lsm.github.io/access_req/JULES_access.html. The infrastructure for building and running UM-JULES simulations uses the Rose suite engine (https://metomi.github.io/rose/doc/html/index.html) and scheduling using the Cylc

work flow engine (https://cylc.github.io/, Oliver et al., 2019). Both Rose and Cylc are available as part of version 3 of the GNU General Public License. The BARRA-C Rose/Cylc suite, with an identifier u-ak499, is version-controlled under the Met Office Science Repository Service and contains the UM-JULES science namelist and simulation configurations. Output from the model simulations was converted from UM fieldsfile format to netCDF4 format using Iris (https://scitools.org.uk/iris/docs/latest/).


**Data availability**

The BARRA datasets for the period of January 1990 to February 2019 are available for academic use. Readers are referred to http://www.bom.gov.au/research/projects/reanalysis (last access: 31 August 2020; Bureau of Meteorology, 2020) for



information on available parameters, access and licensing. The BARRA-R datasets used to initialise and constrain BARRA-C at the boundaries and the BARRA-C ancillary files can be requested by contacting the authors directly and are subjected to the same licensing conditions.

**Author contributions**

PS, DJ, PFH and CJW conceived and/or designed BARRA. NE, CHS and PS developed the BARRA-C system with inputs from CF. NE performed the production, and CHS and NE the evaluation. CHS and NE prepared the paper with contributions from all co-authors.

**Acknowledgments**

Funding for this work was provided by emergency service agencies (New South Wales Rural Fire Service, Western Australia Department of Fire and Emergency Services, South Australia Country Fire Service, South Australia Department of Environment, Water and National Resources) and research institutions (Antarctic Climate and Ecosystems Cooperative Research Centre (ACE CRC) and the University of Tasmania). Funding from Tasmania is supported by the Tasmanian Government and the Australian Government, provided under the Tasmanian Bushfire Mitigation Grants Program.

BARRA-C is set up with assistance from the UKMO colleagues (Stuart Webster) and many colleagues at the Bureau of Meteorology (Greg Kociuba, Gary Dietachmayer, Hongyan Zhu, Yimin Ma, Ilia Bermous, Robin Bowen), the Commonwealth Scientific and Industrial Research Organisation (CSIRO; Martin Dix), and National Computational Infrastructure (NCI; Dale Roberts, Grant Ward). The FSS analysis with the Rainfields2 product is undertaken with assistance from Susan Rennie, Kevin Cheong, and Alan Seed (Bureau of Meteorology), and Suwash Acharya (University of Melbourne). We also thank Mitchell Black and Vinodkumar for their feedback on drafts of the paper. The BARRA project was undertaken with the assistance of resources and services from NCI, which is supported by the Australian Government. This study uses the ERA-Interim and ERA5 data provided through the ARC Centre of Excellence for Climate System Science (Paola Petrelli) at NCI.

ERA-Interim can be retrieved from the ECMWF at https://www.ecmwf.int/en/forecasts/datasets/archive-datasets/reanalysis-datasets/era-interim. ERA5 can be retrieved from Copernicus Climate Data Store, at https://cds.climate.copernicus.eu/. The AWAP data can be requested from http://www.bom.gov.au/climate (last access: 31 August 2020). The Rainfields2 radar product is retrieved from the Rainfields Archiving System provided by the Bureau of Meteorology.





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



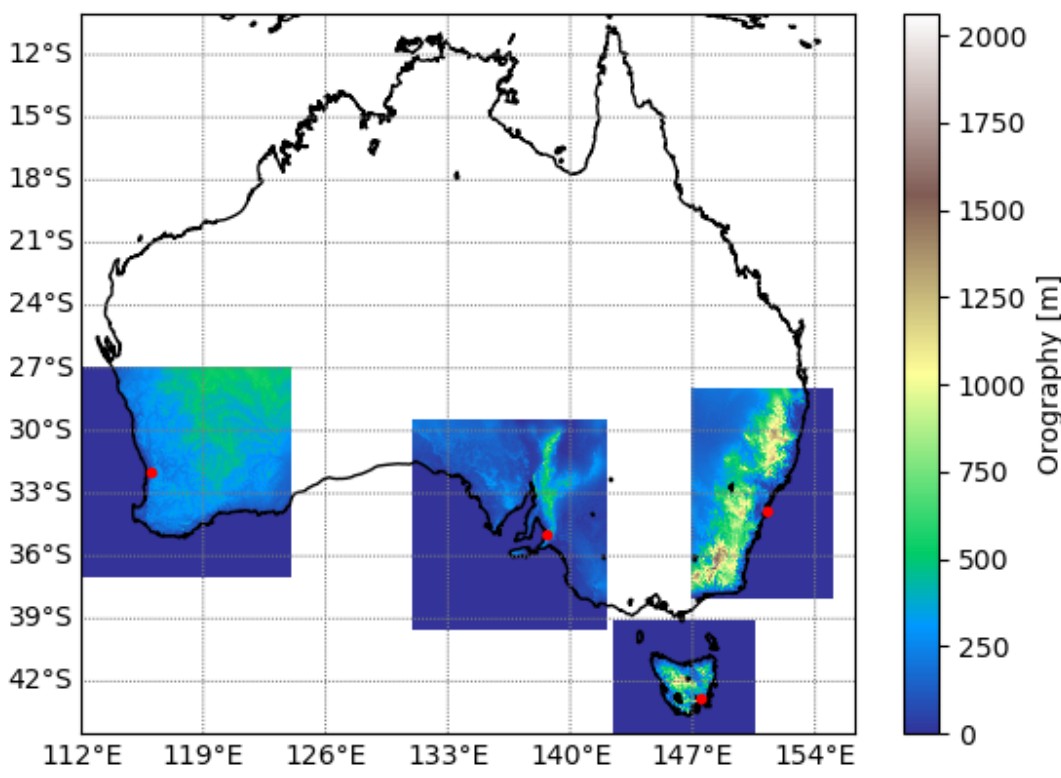

Figure 1: Domains of BARRA-C, (left to right) BARRA-PH (over Perth), BARRA-AD (Adelaide), BARRA-TA (Tasmania), and BARRA-SY (Sydney), showing the modelled orography. Red dots indicate the state capital cities.

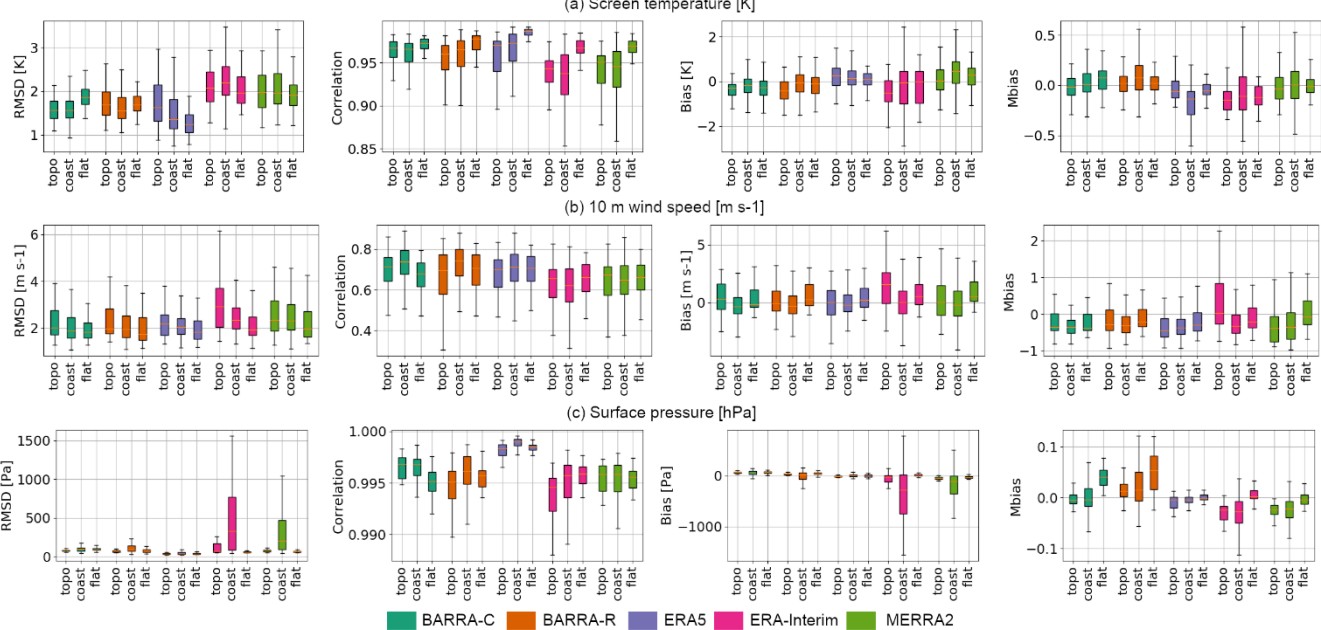

**Figure 2: Box plots showing the distribution of evaluation scores of various models for (a) screen-level temperature, (b) 10 m wind speed, and (c) surface pressure across the four BARRA-C domains. Three regions are analysed separately: coastal ('coast'), complex topography ('topo'), and flat, and the models distinguished by colours. The scores are calculated on model hindcasts valid between 05-07 UTC, and 17-19 UTC against observations during 2010–2012.**




**Figure 3: Mean difference in (a) summer (DJF) daily maximum temperature, (b) winter (JJA) daily minimum temperature and (c)**
**number of days with temperature exceeding 35 ⁰C, in various models during 1990-2018, with respect to AWAP. The models are**
**regridded onto the AWAP grid with the nearest-neighbour interpolation.**



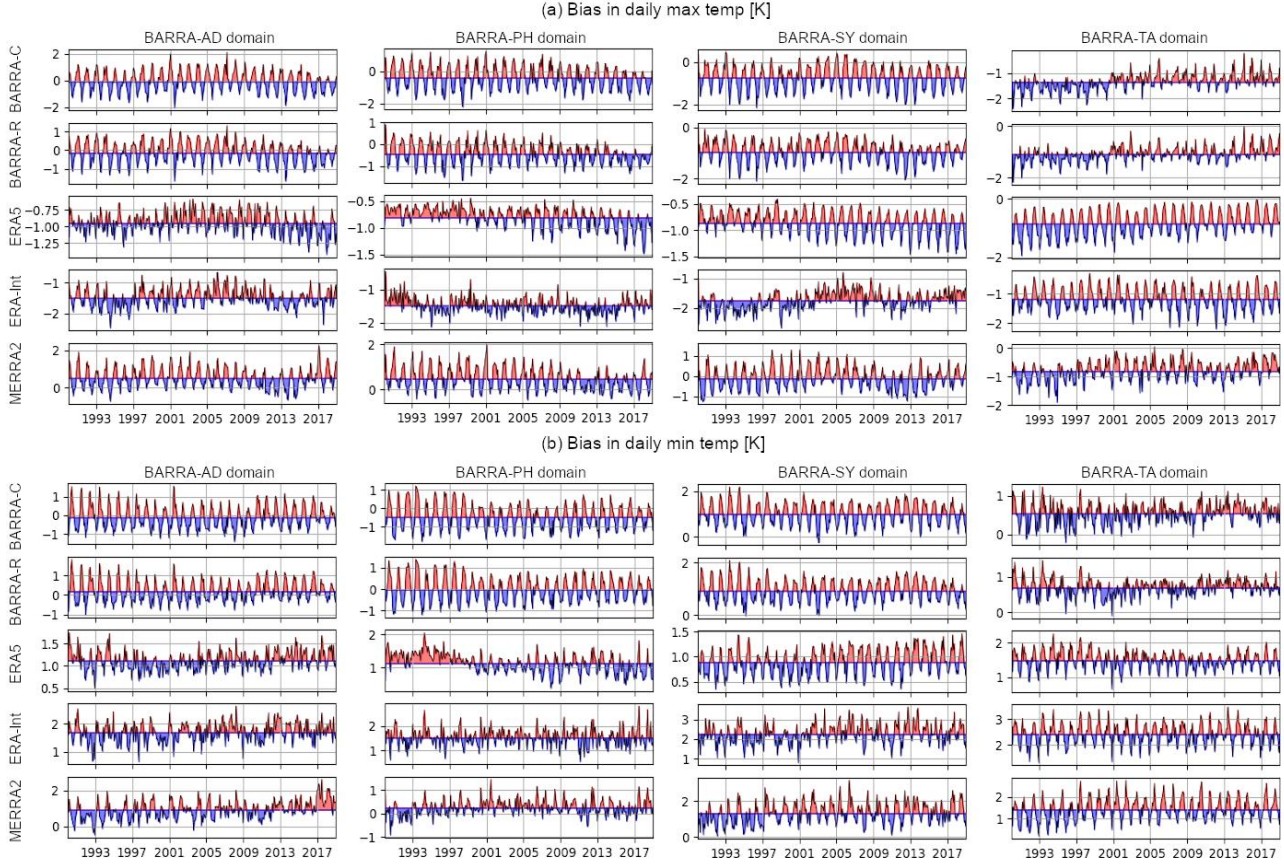

**Figure 4: Monthly mean difference in daily maximum temperature averaged over various BARRA-C domains, with respect to AWAP. Black curves are shaded around the 1990-2018 means. Note that the y-axes are different.**





**Figure 5: Frequency distributions of daily maximum temperature at four AWAP grid points near the major cities in each domains, namely Adelaide (34.92$^0$S, 138.62$^0$E), Perth (31.92$^0$S,115.97$^0$E), Sydney (33.86$^0$S,151.20$^0$E) and Hobart (42.83$^0$S,147.50$^0$E), identified in Figure 1.**






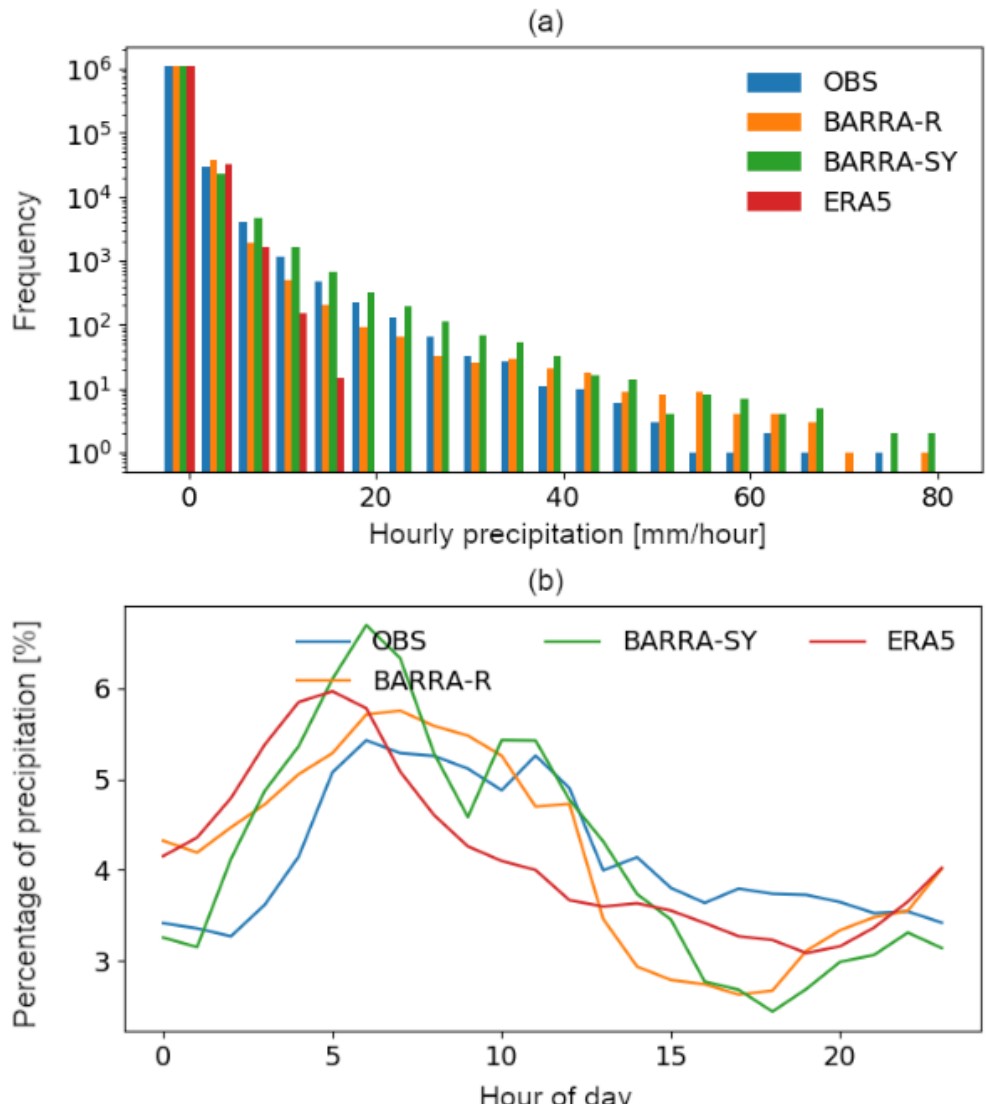

**Figure 6: Distribution of (a) hourly rain rate (mm/h) and (b) rain over 24 hours in UTC, over Sydney during November to February of 2006-2018.**







**Figure 7: Mean difference in (a) annual precipitation and (b) annual count of wet days with depth ≥ 10 mm. The models are regridded onto the AWAP grid with the nearest-neighbour interpolation.**





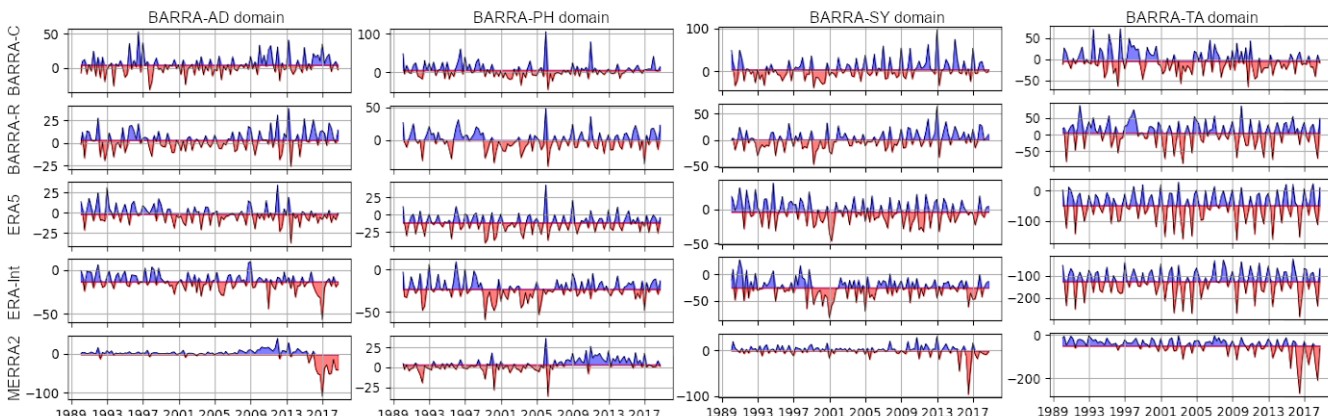

**Figure 8: Mean difference in seasonal precipitation totals over various BARRA-C domains, with respect to AWAP. Black curves are shaded around the 1990-2018 means. Note that the y-axes are different.**

Figure 9: Simulated 6-hour rainfall accumulation [mm] in BARRA-SY and BARRA-R, compared with rainfall derived from the radar network in the Sydney area for five events.






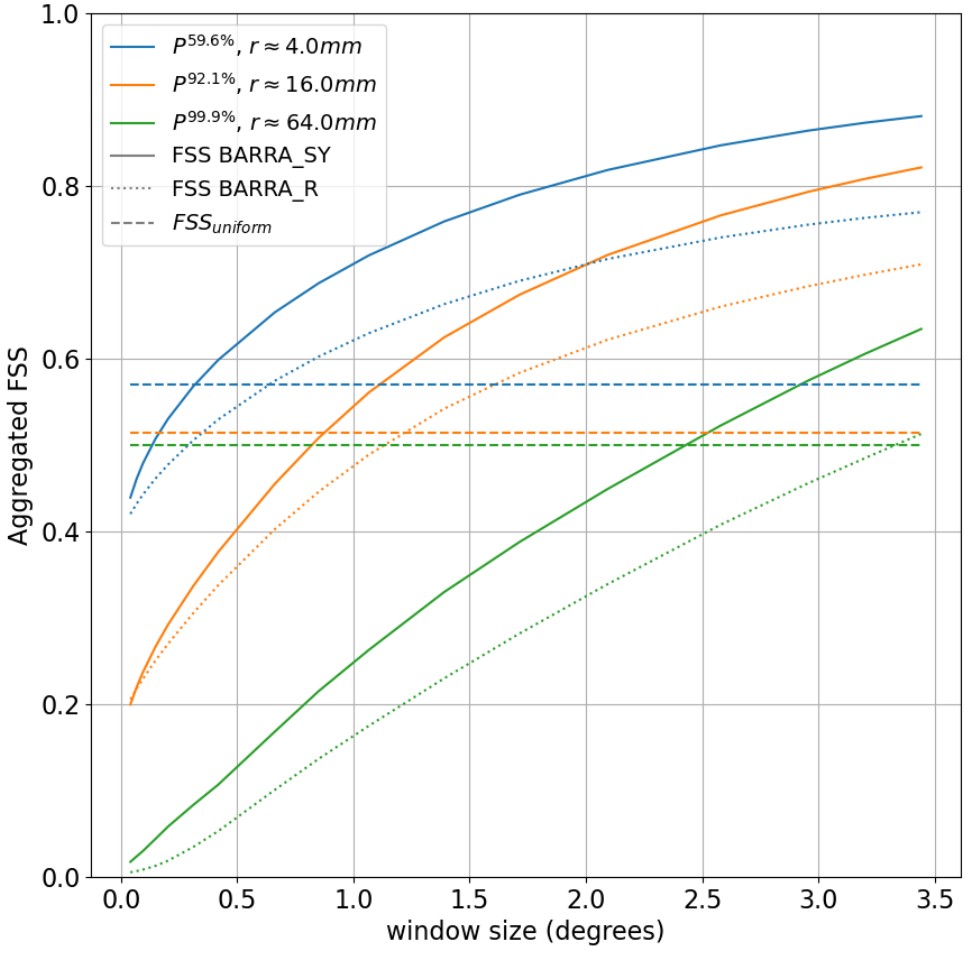

**Figure 10: Aggregated FSS across 1323 6-hour storm events as a function of neighbourhood distance (degrees) for 6h rainfall above three percentile thresholds (distinguished by colours, percentile values, and observed amount in mm). The solid curves indicate the score for BARRA-SY, dotted curves for BARRA-R, and the dashed horizontal lines the uniform score (FSS$_{uniform}$) for each threshold as specified by Roberts and Lean (2007). All 1323 6-hour events have at observations matching the 59$^{th}$ and 92$^{nd}$**

**threshold values, however only 97% of BARRA-SY events matched the 99.9$^{th}$ threshold and only 81% of BARRA-R events.**




**Table 1: Major differences between BARRA-C, BARRA-R and RAL1-M models. The configurations for BARRA-R are described in Su et al. (2019) and Walters et al. (2017), and those for RAL1-M in Bush et al. (2020).**

| Aspects | BARRA-R | BARRA-C | RAL1-M |
|---|---|---|---|
| Nesting setup | Nested in 6-hourly ERA-Interim boundary conditions | Nested in hourly BARRA-R boundary conditions | NA |
| Horizontal grid length in radial resolution | 0.11° | 0.0135° | 0.0135 to 0.04° |
| Vertical model level set | 70 levels, with 50 levels below 18 km, and 20 levels above this, fixed model lid of 80 km above the sea level. | 70 levels, with 61 levels below 18 km, 9 levels above this, fixed model lid of 40 km above sea level | |
| Model timestep | 300 seconds | 60 seconds | 60-100 seconds, depending on the model resolution |
| UM model version | 10.2 | 10.6 | ≥ 10.6 |
| JULES model version | 3.0 | 4.7 | ≥ 4.8 |
| Data assimilation | 6-hourly 4D variational analysis | None | NA |
| Moisture variable SL advection schemes | Quasi-monotone (Bermejo and Staniforth, 1992) | | Posteriori monotonicity filter (PMF) |
| Convective parameterization scheme | Mass-flux convection scheme of Gregory and Rowntree (1990) | None | |
| Gaseous absorption (radiation) scheme | GA6 (Walters et al., 2017) | | GA7 (Walters et al., 2019) |
| Include spectral land-surface albedo | No | | Yes |
| Canopy radiation back-scatter scheme | Isotropic | | Anisotropic |
| Cloud microphysics scheme | Single moment scheme based on Wilson and Ballard (1999) | Wilson and Ballard (1999), with prognostic graupel (Wilkinson and Bornemann, 2014) and improved warm rain scheme (Boutle et al., 2014a) | |
| Boundary layer scheme | 1D vertical turbulent mixing scheme of Lock et al. (2000) | Blended boundary layer parameterisation (Boutle et al., 2014b) | |
| Land surface and hydrology | GA6 (Walters et al., 2017), PDM subgrid-scale heterogeneity, JULES urban parameters are optimized for Australia (Dharssi et al., 2015) | | GA7 (Walters et al., 2019) where TOPMODEL is used |
| BL stochastic perturbations | None | Perturbation to temperature | Perturbation to temperature and moisture |
| BL stability functions | For stable BL, the "sharp" function of Lock et al. (2016) is used over the sea, and over land is a blended combination of the Louis (1979) and the "sharpest" function, for heights below 200 m. The convective BL stability functions are based on UKMO Large-Eddy model simulations. | The "sharpest" function for stable BL everywhere. The convective BL stability functions are based on UKMO Large-Eddy model simulations. | |
| Critical relative humidity profile | 0.92 in the lowest layer, with a gradual decrease to 0.8 at model level 17 (~2100 km). | 0.96 in the lowest layer, and decrease to 0.8 at model level 15 (~850 km). | |