# Peer review of "BARRA v1.0: Kilometre-scale downscaling of an Australian regional atmospheric reanalysis over four midlatitude domains"

_Geoscientific Model Development, 2020_

## Referee Comment (RC1) · Anonymous Referee #1 · 12 Jan 2021

This paper is an evaluation of BARRA v1.0 reanalysis. The authors compare BARRA-C, a 1.5 km downscaled re-analysis against various observations, and investigate added value to BARRA-R, which is the 12 km continental version.

This is a highly valuable paper and generally very well written. I recommend acceptance in GMD subject to the following revisions:

1. The paper would benefit by having an "Observational data-sets" section or similar, where all observations used are described in a bit of detail. Currently details of observations are only provided when results are shown. This would fit well in section 3 of the paper. More details are needed about the point observations and gridded analy-

ses. Could you also provide some background on MERRA2? Why do you choose to use MERRA2 specifically? Some context is needed here. Details about AWAP also needed.

2. Figure 3 – comparisons with AWAP. Interpolation errors within AWAP are available as "RMSE Analysis" from BOM's web-page. It would be useful to show or talk about this to put the biases in context of errors in AWAP?

3. Section 3.2, Figure 3 – you do not discuss the very large bias in Daily max temp in South Australia, which is up to 7.6 degrees C, in the north-west part of the domain. This is a very large bias in both BARRA-C and BARRA-R, and this needs some more attention. MERRA2 has a similar bias in south Australia to BARRA-C and BARRA-R.

4. Figures 3, 4 and 7 – Some analysis of trend would also be useful? Rather than just means. Also, what about variability? It may be useful to examine the standard deviation from AWAP versus the models?

5. Figure 4 – this is a very busy figure, and I urge the authors to find better ways to summarize/plot the data. It is very hard to compare with the different y-axis limits. I suggest to have fewer plots, 1 plot per domain, for tmax and tmin separately. Don't plot the difference, but plot AWAP as solid black line, and each model for that domain as a different color/marker. This is currently too much to digest. By plotting on the same plot, with similar axis, one can actually makes sense of all of this. This Figure is much too busy to digest. Reduce the number of plots, it is very common to plot up to 4-5 lines on a single plot.

6. Figure 5 – could you make the models dotted lines do they are easier to distinguish from AWAP in solid blue. Line 270 – BARRA-R and BARAA-C have too many warm days in Perth, but also, all other models seem to have too few, or close to AWAP. For Adelaide in SA, it seems that BARRA-C does worst than BARRA-R for warm extremes?

7. Figure 8 and related text – same comment as for Figure 4.

[Figure]

8. Can you come up with some objective measure of added value of BARRA-C over BARRA-R? There are many metrics used to quantify added value. There is a lot of literature on quantifying added value of downscaling GCMs using RCMs for example. The same concepts could be applied to quantify added value of BARRA-C over BARRA-R. Producing re-analysis at 1.5 km resolution, takes a lot of effort and the data storage is difficult, as I am sure the authors would know very well. Quantifying the added value would be useful I think.

9. The discussion and conclusion mostly speculates for reasons to explain model biases, that is to be expected. But the paper would be much more interesting if some dynamical analysis were carried out to better understand differences between the models. This could be comparing MSLP patters during very hot extreme events, just as one example. I think this would make the paper much more interesting. I would like the authors to think a bit more about this. The paper would benefit from some more actual analysis of model dynamics. You cannot do everything, I understand, but there is room for the basic dynamics analysis I think.

---

## Referee Comment (RC2) · Anonymous Referee #2 · 10 Feb 2021

The "BARRA v1.0: Kilometre-scale downscaling of an Australian regional atmospheric reanalysis over four midlatitude domains" paper evaluates the performance of high resolution downscaled regional reanalysis datasets over four Australian cities, collectively BARRA-C, derived from the coarser resolution continent-spanning BARRA-R dataset.

The paper rightly points out the relative lack of very high resolution, in this case, km-scale, reanalysis datasets over Australia (and the Australasian region in general) and goes on to evaluate the four BARRA-C domains against observational datasets on annual and seasonal timescales and for case study events, acknowledging the inherent biases often present in these km-scale models.

[Figure]

Recommend acceptance for publication in GMD with consideration for the following comments and revisions (mostly around readability):

1. Throughout the text the phrase "km-scale" and "kilometre-scale" are interchanged. Advise picking one form and sticking with that. 2. Section 1, Line 30: consider changing "over global models" to "beyond current global models" and highlight some of these new insights 3. Section 1, Line 33: change "four km" to "four kilometres" 4. Section 1, Line 55: remove comma in "instruments, to form" 5. Section 1, Line 63: consider changing "Higher resolution reanalyses is needed" to "Such scales are needed" 6. Section 1, Line 72: consider changing "Germany with assimilation of conventional" to "Germany, assimilating conventional" 7. Section 1, Line 72: consider changing "rain rates and demonstrated improved" to "rain rates, demonstrating improved" 8. Section 1, Line 76: consider changing "scale finer than coarser-resolution" to "scale finer than those of coarser resolution" 9. Section 1, Line 87: consider changing "10 minutes" to "10 minute" 10. Section 2, Line 103: change " includes 4 " to "includes four" 11. Section 2, Line 109: consider changing "to hot summer" to "to hot summers" and "differs with cooler" to "differs with a cooler" 12. Section 2.1, Line 124: consider changing "model in BARRA-C" to "model used in BARRA-C" and remove comma after "time step" 13. Section 2.1 details differences between the BARRA-R and BARRA-C model configurations and compares these to the RAL1 science configuration, now available in the Unified Model. Given neither BARRA-R nor BARRA-C uses RAL1 as the core science configuration, not convinced this adds much to the discussion about the performance of BARRA-C, other than to point out a newer science configuration is available for use. This discussion might make more sense if some BARRA-C runs had been done using the RAL1 configuration and comparative results presented. Otherwise would be consider re-wording to describe the OS36 science configuration used by BARRA-C. 14. Section 2.2, Line 183: consider changing "benefits of analysis in BARRA-R is inherited" to "benefits of the BARRA-R analysis is inherited" 15. Section 2.2, Line 190: consider changing "to the UM grid" to "to the BARRA-C UM grids." 16. Section 2.2, Line 192: consider changing "In other words, the" to "Therefore the" 17. Section 2.2,

Line 193: consider changing "considered short, and is chosen to meet computational constraints and regular reinitialisation is needed" to "considered short, but is chosen to meet computational constraints with regular reinitialisation needed" 18. Section 3.2 As part of the discussion comparing BARRA against AWAP observations, the authors begin to discuss the impact that choosing the closest model grid cell to an observation location can have on validation results, especially if the closest model grid point is of a different land use type of not even a land point at all. At this point, it would be good to see in a bit more detail about the datasets used to define these characteristics and, at least for one observation site, how different these land cover fractions or land use types are across the BARRA models. Can this assertion be backed up? Could be added to the Supplemental material. 19. Section 3.4, Line 299: consider changing "in all but TA domains." to "in all the TA domain." 20. Section 3.5, Line 318: consider changing "blended with the gauge" to "blended with gauge" 21. In relation to Figure 9(v), despite being orographically tied, is there anything that stands out about the storm case that might lead to BARRA-R rainfall performing markedly different from BARRA-C in this case, and from the other BARRA-R storm events shown? 22. Section 4, Line 349: consider changing "positive (negative) bias during light (strong)" to "positive (negative) during light (strong)" 23. Section 4, Line 354: change "reference therein" to "references therein" 24. Section 4, Line 365: be consistent with use of "z" or "s" in parameterization/parameterisation throughout the document 25. Section 4, Line 368: consider changing "in other studies; Lean et al. (2008) and Hanley et al. (2016) found that" to "in other studies. For example, Lean et al. (2008) and Hanley et al. (2016) found that" 26. Section 5, Line 417: consider changing "pressure-level grids from BARRA-C" to "pressure-level gridded data from BARRA-C" 27. Section 5, Line 427: consider changing "albeit high rainfall bias exists." to "albeit with high rainfall bias." 28. Line 441: consider changing "UM is available" to "The UM is available" 29. Line 445: consider changing "available as part of the version 3" to "available under Version 3" 30. Line 455: consider changing "are subjected to" to "are subject to" 31. For general consistency across the document should decide whether all URLs are to

be hyperlinks or plain text. Currently have a mix of both formats. 32. Figure 1: for clarity, worth changing last sentence to "Red dots indicate the location of the state capital cities." 33. Figures 2, 4 and 8 are quite hard to read as printed. Wonder if these would benefit from changing to a landscape orientation. In Figure 8 especially, it is hard to see the black curves indicated in the caption and these means are not mentioned in the manuscript otherwise that I noticed. 34. Figures 3 and 7: consider changing "AWAP grid with the nearest neighbour" to "AWAP grid using the nearest neighbour" 35. Figure 5: consider changing "in each domain" to "in each BARRA-C domain". Also remove/add extra spaces in the coordinates of each city for consistency. 36. Figure 10: change "6h" to 6-hour" for consistency with rest of the document. Wonder if the last sentence in the caption is best moved to the main text of the manuscript as part of the main discussion (Section 3.5) rather than left in the Figure caption.

---

## Author Comment (AC1) · 31 Mar 2021

**1. Response to general comments from Reviewer R1**

*This paper is an evaluation of BARRA v1.0 reanalysis. The authors compare BARRAC, a 1.5 km downscaled re-analysis against various observations, and investigate added value to BARRA-R, which is the 12 km continental version. This is a highly valuable paper and generally very well written. I recommend acceptance in GMD subject to the following revisions.*

We would like to thank the reviewer for their thorough and constructive review. We respond to the comments, in turn below, which we believe has led to an improved paper.

**2. Response to specific questions from Reviewer R1**

1. *The paper would benefit by having an "Observational data-sets" section or similar, where all observations used are described in a bit of detail. Currently details of observations are only provided when results are shown. This would fit well in section 3 of the paper. More details are needed about the point observations and gridded analyses. Could you also provide some background on MERRA2? Why do you choose to use MERRA2 specifically? Some context is needed here. Details about AWAP also needed.*

We agree that more information can be provided for the various reference data sets. We have considered adding another section or subsection within or before Sec. 3, but find this disrupts the flow of the paper. We will add a section S1 in the Supplement to include (1) Table that summarizes the characteristics of the reference data sets (including spatiotemporal resolutions, references and parameters we used), (2) details and issues of AWAP and Rainfields2, and (3) distinctions between the global reanalyses.

In Sec. 3., we will refer readers to Sec. S1 of the Supplement, and add a comment "To increase the diversity of models used in our inter-comparison, we also include the Modern-Era Retrospective analysis for Research and Applications-2 (MERRA2, Gelaro et al., 2017) hindcasts.". We find that we need three distinctive systems to distinguish model biases in view of limitations to observational data sets; here we have BARRA-R with UM and 4DVar, ERA reanalyses with IFS and 4DVar, and MERRA2 with GEOS and 3DVar.

2. *Figure 3 – comparisons with AWAP. Interpolation errors within AWAP are available as "RMSE Analysis" from BOM's web-page. It would be useful to show or talk about this to put the biases in context of errors in AWAP?*

We agree. A section S1 will be added to the Supplement to discuss issues of AWAP. In particular for AWAP, S1 notes
"*AWAP provides gridded daily 0.05 × 0.05° analysis of station observed maximum and minimum 2 m temperature data, and raingauge-based daily accumulation of precipitation. The grids for temperature are generated using an optimized Barnes successive-correction method that applies weighted averaging to the station data. Topographical information is included by using anomalies from long-term (monthly) averages in the analysis process. By contrast, the ratio of observed rainfall to monthly average is used in the analysis process of precipitation. Readers are referred to Jones et al. (2009) for details. The root-mean-squared error (RMSD) of the daily maximum temperature analysis is around 1-1.5 K over the BARRA-C domains. The error is higher around Nullarbor Plain (northwest of the BARRA-AD domain) due to a relatively sparse network, and the Southeastern highlands (BARRA-SY domain) due to strong temperature gradients between the coast and mountains. The analysis errors are larger for daily minimum temperature than for daily maximum values, with RMSD around 1.5-2 K and larger errors in the regions. For daily precipitation analysis, the RMSD over the BARRA-C domains is relatively uniform at around 2.5 mm. Higher RMSD of 5 mm*

*is noted over northwestern coastal regions of BARRA-SY domain. Further, Chubb et al. (2016) has shown that the AWAP error for wintertime precipitation over the Snowy Mountains (BARRA-SY domain) can be as high as 4.5 mm, due to the lack of gauges and steep topography exposed to prevailing winds. At high elevations where frozen precipitation is challenging to measure, AWAP analysis has underestimated the total precipitation amount by more than 10%. Therefore, the comparisons with AWAP for the Southeastern Highlands and the Nullarbor Plain need to be interpreted in view of these limitations. King et al. (2012) has also found AWAP tends to report lower extreme rainfall estimates (e.g., climatological 95th percentile rainfall) than those observed at stations, which is characteristic of an interpolated product."*

This may partly explain the warm bias observed over the Nullarbor Plain for daily maximum temperature in Figure 3. Apart from this, it is difficult to definitively attribute biases in various reanalyses to error patterns of AWAP.

3. *Section 3.2, Figure 3 – you do not discuss the very large bias in Daily max temp in South Australia, which is up to 7.6 degrees C, in the north-west part of the domain. This is a very large bias in both BARRA-C and BARRA-R, and this needs some more attention. MERRA2 has a similar bias in south Australia to BARRA-C and BARRA-R.*

We agree. BARRA, MERRA2 and ERA5 all show warm bias, with respect to AWAP, over the Nullarbor Plain. This region has very few observing stations. According to [http://www.bom.gov.au/climate/data/](http://www.bom.gov.au/climate/data/), there are only around 5 stations reporting daily maximum temperature over this 2x2 degree region. Differences in the land cover classification between BARRA and ERA reanalyses may account for some of their differences; BARRA is based on IGBP while ERA is likely based on CCI land cover. The differences in the land cover classification between BARRA and ERA5 appear to contribute to the differences in temperature bias seen over the salt lakes in the AD domain. To investigate this further, work is in progress to map CCI raw data to the local land cover types used by UM; the current transform table has led to large differences between IGBP and CCI in terms of shrub and bare soil covers in this region. These comments will be added to Sec. 3.2, and with reference to the Supplement where the quality of AWAP is noted.

4. *Figures 3, 4 and 7 – Some analysis of trend would also be useful? Rather than just means. Also, what about variability? It may be useful to examine the standard deviation from AWAP versus the models?*

We find that the trends are only apparent in few models and for few domains (noted in text), and are generally not systematic across time period, across the models and domains to warrant trend analysis. Further, comments on possible trend of the temperature max/min bias for these cases are made in Sec. 3.2 and 3.4.

We will extend the bias analysis to compare their standard deviation of daily min/max temperature in each month and present the results in the Supplement. Section 3.2 will be extended with the following observations of this additional analysis:

*"This analysis of variability of bias is also repeated for the standard deviation of the modelled temperature and AWAP in Figure S4 and S5 of the Supplement. BARRA-C shows a slightly wider dispersion of daily maximum temperature than AWAP (by 0.4 K) and BARRA-R (by 0.1 K), with the exception for the TA domain. For BARRA-TA, the standard deviation of BARRA is similar to AWAP and is higher than the global reanalyses. For daily minimum temperature, both BARRA are similar and they are generally under-dispersed by 0.3 K compared to AWAP.*

[Figure]

**Figure S4: As with Figure 4, but for monthly difference in standard deviation of daily maximum temperature over various BARRA-C domains, with respect to AWAP. The timeseries are shaded around their individual 1990-2018 means.**

[Figure]

**Figure S5: As with Figure S4, but for daily minimum temperature.**

"

5. *Figure 4 – this is a very busy figure, and I urge the authors to find better ways to summarize/plot the data. It is very hard to compare with the different y-axis limits. I suggest to have fewer plots, 1 plot per domain, for tmax and tmin separately. Don't plot the difference, but plot AWAP as solid black line, and each model for that domain as a different color/marker. This is currently too much to digest. By plotting on the same plot, with similar axis, one can actually makes sense of all of this. This Figure is much too busy to digest. Reduce the number of plots, it is very common to plot up to 4-5 lines on a single plot.*

Figure 4 aims to demonstrate the seasonal and inter-annual variations in the model biases (w.r.t. AWAP). As the bias is in the order of 1-2K, revising the plots as per the reviewer's suggestion will not capture this. We agree that the figure is currently small and busy but is so in order to capture differences between the domains, different reanalyses, and between daily maximum and minimum temperatures. For the readability of the subfigures, we will split the figure into two sets of subfigures, one for daily maximum temperature (Figure 4), and another for daily minimum temperature (Figure 5).

6. *Figure 5 – could you make the models dotted lines do they are easier to distinguish from AWAP in solid blue. Line 270 – BARRA-R and BARAA-C have too many warm days in Perth, but also, all other models seem to have too few, or close to AWAP. For Adelaide in SA, it seems that BARRA-C does worst than BARRA-R for warm extremes?*

We will improve Figure 5 by changing the plotting line styles. In particular, we expand this with the added value analysis (see reply to comment R1#8) to look at warm extremes across the domains, not limiting to few selected points. We move Figure 5 and this discussion to the Supplement.

7. *Figure 8 and related text – same comment as for Figure 4.*

See our reply to R1#4.

8. *Can you come up with some objective measure of added value of BARRA-C over BARRA-R? There are many metrics used to quantify added value. There is a lot of literature on quantifying added value of downscaling GCMs using RCMs for example. The same concepts could be applied to quantify added value of BARRA-C over BARRA-R. Producing re-analysis at 1.5 km resolution, takes a lot of effort and the data storage is difficult, as I am sure the authors would know very well. Quantifying the added value would be useful I think.*

In response to this comment, we report an added value analysis in a new Sec. 3.6, which reads, "

**3.6 Added value analysis for temperature and rainfall extremes**

*We apply an approach similar to Di Luca et al. (2015) to quantify the added value (AV) in the representation of climatological extremes from BARRA-C by comparing the skill between the BARRA-C and BARRA-R. The warm extremes of daily maximum temperature, the cold extremes of daily minimum temperature and the wet extremes of daily precipitation are assessed against AWAP, noting that the true AV from BARRA-C at its native resolution is not fully determined here. The statistics for extremes (X) are given by the percentiles of the daily temperature and precipitation values over the 29-year time period. We use $AV_d=[d(X_{(BARRA-R)},X_{AWAP})-d(X_{(BARRA-C)},X_{AWAP})]/[d(X_{(BARRA-R)},X_{AWAP})+d(X_{(BARRA-C)},X_{AWAP})]$ of Di Luca et al. (2016) where d defines a distance metric between the model-derived and AWAP-derived statistics computed across the grid cells. To capture both the total errors and spatial patterns of the statistics, we let $d=MSE(A,B)=E[(A-B)^2]$ defining the mean squared error and $d=Corr(A,B)=1-R(A,B)$, where $E(∘)$ is the expectation operator and R their Pearson's correlation. Larger positive AV values suggest smaller errors in BARRA-C than in BARRA-R and thus substantial added value by the downscaling of BARRA-R.*

*Figure 11 plots AVs for different BARRA-C domains, showing that AV is not gained consistently across the percentiles, variables and domains. For warm extremes of daily maximum temperature, BARRA-C shows positive AVMSE over BARRA-R in the TA and AD domains. However, there are low or negative AVMSE for AD, PH and SY (inland region) mainly due to the warm bias in BARRA-C, also seen in Figure 3(c) and 6(a,b). With positive AVCorr, BARRA-C captures the spatial patterns of the warm extremes across the domains, particularly over the coastal and high topography regions (Figure S6 of the Supplement). For cold extremes in Figure 11(b), BARRA-C still shows positive AVMSE over all but the SY domain, due to AV over the coastal regions. The negative AVMSE in SY is related to warmer cold extremes, particularly over the Great Dividing Range. Positive AVCorr is seen in TA but not in the other domains, although it should be noted that the BARRAs are generally strongly correlated with AWAP with R mostly between 0.7 to 0.9.*

*AV from BARRA-C for wet extremes of precipitation relates more to the spatial patterns of the extremes (Figure 11(c)). Given the tendency of BARRA-C to overestimate heavy rainfall, the wet bias relative to AWAP, particularly over the PH domains (Figure 7(b)), is responsible for the low AVMSE. For the SY domain, positive AVCorr for precipitation agrees with the above FSS analysis, which somewhat avoids the issue of bias through percentile-based thresholding. Furthermore, AVCorr is positive for extreme rainfall for all but the AD domain indicating that despite the wet bias, rainfall extremes in BARRA-C can be better spatially correlated with AWAP. Assessing AV for wet extremes may also be problematic with AWAP. As an interpolated dataset, AWAP tends to underestimate the intensity of extreme heavy rainfall observed at stations and the issue is more*

*pronounced at locations with sparse observational sampling or high topography, particularly in SY and TA (Chubb et al. 2016; King et al., 2012).*

[Figure]

**Figure 11: Added value (AV) analysis of the (a) warm extreme of daily maximum temperature, (b) cold extreme of daily minimum temperature, and (c) wet extreme of daily precipitation, performed for different BARRA-C domains.**

"

9. *The discussion and conclusion mostly speculates for reasons to explain model biases, that is to be expected. But the paper would be much more interesting if some dynamical analysis were carried out to better understand differences between the models. This could be comparing MSLP patters during very hot extreme events, just as one example. I think this would make the paper much more interesting. I would like the authors to think a bit more about this. The paper would benefit from some more actual analysis of model dynamics. You cannot do everything, I understand, but there is room for the basic dynamics analysis I think.*

This paper is submitted to Geoscientific Model Development journal as a model experiment description paper. It is outside the scope of this work to provide detailed analysis of model dynamics. We have provided references, namely Bush et al. (2020, regional configuration of the Unified Model), Champion and Hodges (2014, model spin up issues), Lean et al. (2008) and Hanley et al. (2016, convection) for some of the model issues also observed in this work.

---

## Author Comment (AC2) · 31 Mar 2021

**1. Response to general comments from Reviewer R2**

*The "BARRA v1.0: Kilometre-scale downscaling of an Australian regional atmospheric reanalysis over four midlatitude domains" paper evaluates the performance of high resolution downscaled regional reanalysis datasets over four Australian cities, collectively BARRA-C, derived from the coarser resolution continent-spanning BARRA-R dataset.*
*The paper rightly points out the relative lack of very high resolution, in this case, kmscale, reanalysis datasets over Australia (and the Australasian region in general) and goes on to evaluate the four BARRA-C domains against observational datasets on annual and seasonal timescales and for case study events, acknowledging the inherent biases often present in these km-scale models.*
*Recommend acceptance for publication in GMD with consideration for the following comments and revisions (mostly around readability).*

We would like to thank the reviewer for their careful review and positive recommendation for this paper. We respond to the comments, in turn below, which we believe have improved our first submission.

**2. Response to specific questions from Reviewer R2**

1. *Throughout the text the phrase "km-scale" and "kilometre-scale" are interchanged. Advise picking one form and sticking with that.*

Agree to use "kilometre-scale" consistently.

2. *Section 1, Line 30: consider changing "over global models" to "beyond current global models" and highlight some of these new insights*

Agree to make the textual change with the revised text:
*"Similarly, CPMs have provided new insights in regional climate projections (e.g., Argüeso et al., 2014; Prein et al., 2015; Kendon et al., 2017; 2019) beyond current global models. For instance, regional CPMs have suggested that future increases in short-duration precipitation extremes are larger than what can be expected from increases in atmospheric moisture alone (Kendon et al., 2021 and references therein). Major efforts are underway toward refining the horizontal resolution of global climate models to kilometre-scale (Schär et al., 2020)."*.

3. *Section 1, Line 33: change "four km" to "four kilometres"*

Agree.

4. *Section 1, Line 55: remove comma in "instruments, to form"*

Agree.

5. *Section 1, Line 63: consider changing "Higher resolution reanalyses is needed" to "Such scales are needed"*

Agree.

6. *Section 1, Line 72: consider changing "Germany with assimilation of conventional" to "Germany, assimilating conventional"*

Agree.

7. *Section 1, Line 72: consider changing "rain rates and demonstrated improved" to "rain rates, demonstrating improved"*

Agree.

8. *Section 1, Line 76: consider changing "scale finer than coarser-resolution" to "scale finer than those of coarser resolution"*

Agree.

9. *Section 1, Line 87: consider changing "10 minutes" to "10 minute"*

Agee.

10. *Section 2, Line 103: change " includes 4 " to "includes four"*

Agree.

11. *Section 2, Line 109: consider changing "to hot summer" to "to hot summers" and "differs with cooler" to "differs with a cooler"*

Agree.

12. *Section 2.1, Line 124: consider changing "model in BARRA-C" to "model used in BARRA-C" and remove comma after "time step"*

Agree.

13. *Section 2.1 details differences between the BARRA-R and BARRA-C model configurations and compares these to the RAL1 science configuration, now available in the Unified Model. Given neither BARRA-R nor BARRA-C uses RAL1 as the core science configuration, not convinced this adds much to the discussion about the performance of BARRA-C, other than to point out a newer science configuration is available for use. This discussion might make more sense if some BARRA-C runs had been done using the RAL1 configuration and comparative results presented. Otherwise would be consider re-wording to describe the OS36 science configuration used by BARRA-C.*

BARRA-C configurations are presented relative to the RAL1 configurations of Bush et al. (2020) because RAL1 is the first defined and published version of the model in a regional set up. To our best knowledge, the OS36 is not defined in a prior published work, but the closely related version OS37 is used as the baseline in Bush et al. (2020) to assess RAL1. The BARRA-C model includes many of the component configurations as RAL1 but Table 1 identifies their differences. Accordingly, the BARRA-C model can be described without needing to completely rewrite the detailed descriptions found in Bush et al. (2020).

However, we appreciate that the impact of the RAL1 configuration differences should be made clearer. In Sec. 2.1, we expand on this to highlight these impacts for differences in stochastic perturbations of moisture, land surface representation and treatment of gaseous absorption. The added text is

*"The mid-latitude version of RAL1 therefore includes stochastic perturbations of temperature and moisture and relative weak turbulent mixing, to encourage the model fields to be less uniform and help convection to initiate. It is of note that the stochastic perturbations of moisture are absent in BARRA-C, and thus may still suffer from the initiation issue. […] BARRA-C however does not include a set of changes to the representation of the land surface and the canopy radiation model in RAL1, which have shown to improve the issue of damped diurnal cycle in near-surface temperatures. BARRA-C also does not benefit from the improved treatment of gaseous absorption in both long- and short-wave regimes in GA7 and RAL1, which improves interaction with band-by-band aerosol and cloud forcing."*

14. *Section 2.2, Line 183: consider changing "benefits of analysis in BARRA-R is inherited" to "benefits of the BARRA-R analysis is inherited"*

Agree.

15. *Section 2.2, Line 190: consider changing "to the UM grid" to "to the BARRA-C UM grids."*

Agree.

16. *Section 2.2, Line 192: consider changing "In other words, the" to "Therefore the"*

Agree.

17. *Section 2.2, C2 Line 193: consider changing "considered short, and is chosen to meet computational constraints and regular reinitialisation is needed" to "considered short, but is chosen to meet computational constraints with regular reinitialisation needed"*

Agree.

18. *Section 3.2 As part of the discussion comparing BARRA against AWAP observations, the authors begin to discuss the impact that choosing the closest model grid cell to an observation location can have on validation results, especially if the closest model grid point is of a different land use type of not even a land point at all. At this point, it would be good to see in a bit more detail about the datasets used to define these characteristics and, at least for one observation site, how different these land cover fractions or land use types are across the BARRA models. Can this assertion be backed up? Could be added to the Supplemental material.*

In view of the comment R1#8 from the first reviewer on added value analysis, we have expanded this local-scale analysis of extreme temperature to become a part of this analysis in Sec. 3.6. In particular, we expand this with the added value analysis (see reply to comment R1#8) to look at warm extremes, cold extremes and wet extremes across the domains, not limiting to few selected locations. We move Figure 5 and the local-scale analysis to the Supplement, where we can add illustrations of the differences in land fractions, topography and surface cover types for Tasmania as an example to illustrate differences between the models. The Supplement notes,
"

*The closest model grid cells are selected, and due to differences in spatial resolution, not all models treat these cells as land points, and even as a land point, they are treated with different land fractions. These affects how representative they are simulating temperature seen at the local scale. Figure S9 illustrates this for Tasmania, where the associated grid cell land fraction varies between*

*0.55 (BARRA-C) to 0.8 (ERA5) and the elevation varies between around 2 m (BARRA-C) to about 200 m (ERA reanalyses).*

[Figure]

**Figure S9: (top) land mask or fraction of the various models centred at the Hobart Airport (red crosses), Tasmania, (middle) their orography, and (bottom) domain surface cover types for BARRAs.**

*"*

19. *Section 3.4, Line 299: consider changing "in all but TA domains." to "in all the TA domain."*

Disagree. To improve clarity, the text is revised as "*in all the BARRA-C domains but TA.*"

20. *Section 3.5, Line 318: consider changing "blended with the gauge" to "blended with gauge"*

Agree.

21. *In relation to Figure 9(v), despite being orographically tied, is there anything that stands out about the storm case that might lead to BARRA-R rainfall performing markedly different from BARRA-C in this case, and from the other BARRA-R storm events shown?*

The storm case in Figure 9(v) is the only summer case shown (i-v) and was attributed to one of a series of surface troughs on the NSW coast that month and positive temperature anomalies. The role of daytime convection in this case may lead to a more pronounced difference in rainfall between BARRA-R and C models compared to the other cases. The BARRA-C rainfall better matches the Rainfields rainfall accumulations in both amount and spatial pattern. BARRA-R rainfall lacks organisation which can be attributed to the lower resolution and the convective parametrisation scheme. Additionally, the lack of mass conservation in the convective parametrisation used in BARRA-R may have lead to the spurious extreme rainfall amounts in the Blue Mountains area in Figure 9(v,c) (>256mm accumulation totals). We have added a brief note in Section 3.5 to this effect.

22. *Section 4, Line 349: consider changing "positive (negative) bias during light (strong)" to "positive (negative) during light (strong)"*

Agree.

23. *Section 4, Line 354: change "reference therein" to "references therein"*

Agree.

24. *Section 4, Line 365: be consistent with use of "z" or "s" in parameterization/parameterisation throughout the document*

Agree.

25. *Section 4, Line 368: consider changing "in other studies; Lean et al. (2008) and Hanley et al. (2016) found that" to "in other studies. For example, Lean et al. (2008) and Hanley et al. (2016) found that"*

Agree.

26. *Section 5, Line 417: consider changing "pressure-level grids from BARRA-C" to "pressure-level gridded data from BARRA-C"*

Agree.

27. *Section 5, Line 427: consider changing "albeit high rainfall bias exists." to "albeit with high rainfall bias."*

Agree.

28. *Line 441: consider changing "UM is available" to "The UM is available"*

Agree.

29. *Line 445: consider changing "available as part of the version 3" to "available under Version 3"*

Agree.

30. *Line 455: consider changing "are subjected to" to "are subject to"*

Agree.

31. *For general consistency across the document should decide whether all URLs are to be hyperlinks or plain text. Currently have a mix of both formats.*

Agree to remove hyperlinks.

32. *Figure 1: for clarity, worth changing last sentence to "Red dots indicate the location of the state capital cities."*

Agree.

33. *Figures 2, 4 and 8 are quite hard to read as printed. Wonder if these would benefit from changing to a landscape orientation. In Figure 8 especially, it is hard to see the black curves indicated in the caption and these means are not mentioned in the manuscript otherwise that I noticed.*

Agree. Figure 2 will be changed to a landscape orientation. Figure 4 will be split up into two figures, one on daily maximum temperature, and another on daily minimum temperature. The arrangement of Figure 8 will be improved to make the subfigures bigger so that the black curves are more visible.

34. *Figures 3 and 7: consider changing "AWAP grid with the nearest neighbour" to "AWAP grid using the nearest neighbour"*

Agree.

35. *Figure 5: consider changing "in each domain" to "in each BARRA-C domain". Also remove/add extra spaces in the coordinates of each city for consistency.*

Agree.

36. *Figure 10: change "6h" to 6-hour" for consistency with rest of the document. Wonder if the last sentence in the caption is best moved to the main text of the manuscript as part of the main discussion (Section 3.5) rather than left in the Figure caption.*

Agree. We also find the sentence needs to be expanded to explain the details of the thresholding for FSS calculation better. To this end, we move the sentence to the Supplement, and expand this.

---

## Author Response (AR2)

**Response to the general comment from Reviewer R1**

*I thank the authors for addressing the comments. The paper can now be published. I would suggest adding 1 line in the abstract about the added value analysis and what it shows overall.*

We would like to thank the reviewer for their second review of the revised manuscript.

We have amended the abstract to address the reviewer's comment. In particular, the abstract is extended to expand on the finding of the added value analysis and writes, "[…] *As a hindcast-only system, BARRA-C largely inherits the domain-averaged bias pattern from BARRA-R but does produce different climatological extremes for temperature and precipitation. An added value analysis of temperature and precipitation extremes shows that BARRA-C provides additional skill over BARRA-R when compared to gridded observations. The spatial patterns of BARRA-C warm temperature extremes and wet precipitation extremes are more highly correlated with observations. BARRA-C adds value in representation of the spatial pattern of cold extremes over coastal regions but remains biased in terms of magnitude.*"

We have also taken this opportunity to proofread the manuscript and made minor textual changes. No new analysis or finding was added. Some text in Section 3 Results is moved to Section 4 Discussions. The changes are documented in the tracked-change version of the manuscript.